METHODS

# BRASS: Permutation methods for binary traits in genetic association studies with structured samples

Joelle Mbatchou[1,2], Mark Abney[3], Mary Sara McPeek[2,3]*

**1** Regeneron Genetics Center, Tarrytown, New York, United States of America, **2** Department of Statistics, The University of Chicago, Chicago, Illinois, United States of America, **3** Department of Human Genetics, The University of Chicago, Chicago, Illinois, United States of America

* mcpeek@uchicago.edu

**Data Availability Statement:** The genotype and phenotype data used in our analyses are previously published and are publicly available from the Dryad online repository with identifier DOI 10.5061/dryad.266k4. The software used in this manuscript

## Abstract

In genetic association analysis of complex traits, permutation testing can be a valuable tool for assessing significance when the distribution of the test statistic is unknown or not well-approximated. This commonly arises, e.g, in tests of gene-set, pathway or genome-wide significance, or when the statistic is formed by machine learning or data adaptive methods. Existing applications include eQTL mapping, association testing with rare variants, inclusion of admixed individuals in genetic association analysis, and epistasis detection among many others. For genetic association testing in samples with population structure and/or relatedness, use of naive permutation can lead to inflated type 1 error. To address this in quantitative traits, the MVNpermute method was developed. However, for association mapping of a binary trait, the relationship between the mean and variance makes both naive permutation and the MVNpermute method invalid. We propose BRASS, a permutation method for binary traits, for use in association mapping in structured samples. In addition to modeling structure in the sample, BRASS allows for covariates, ascertainment and simultaneous testing of multiple markers, and it accommodates a wide range of test statistics. In simulation studies, we compare BRASS to other permutation and resampling-based methods in a range of scenarios that include population structure, familial relatedness, ascertainment and phenotype model misspecification. In these settings, we demonstrate the superior control of type 1 error by BRASS compared to the other 6 methods considered. We apply BRASS to assess genome-wide significance for association analyses in domestic dog for elbow dysplasia (ED) and idiopathic epilepsy (IE). For both traits we detect previously identified associations, and in addition, for ED, we detect significant association with a SNP on chromosome 35 that was not detected by previous analyses, demonstrating the potential of the method.

## Author summary

To determine whether genetic association with a trait is significant, permutation methods are an attractive and popular approach when analytic methods based on distributional assumptions are not available, e.g., when applying machine learning or data adaptive

is publicly available at https://github.com/joellesophya/BRASS.

**Funding:** This project was funded by National Human Genome Research Institute, NIH grant (R01 HG001645 to MSM). The funders had no role in study design, data collection and analysis, decision to publish, or preparation of the manuscript.

**Competing interests:** The authors have declared that no competing interests exist.

methods, or when performing a multiple testing correction, e.g., to assess region-wide or genome-wide significance in association mapping studies. Existing applications include eQTL mapping, association testing with rare variants, inclusion of admixed individuals in genetic association analysis, and detection of genetic interaction among many others. However, when there is population structure in the sample, naive permutation of the data can lead to inflated significance of the association results. For continuous traits, linear mixed-model based approaches have been proposed for permutation-based tests that can also adjust for sample structure; however, these do not remain valid when applied to binary traits, as key features of binary data are not well accounted for. We propose BRASS, a permutation-based testing method for binary data that incorporates important characteristics of binary data in the trait model, can accommodate relevant covariates and ascertainment, and adjusts for the presence of structure in the sample. In simulations, we demonstrate the superior control of type 1 error by BRASS compared to other methods, and we apply BRASS in the context of correcting for multiple testing in two genome-wide association studies in domestic dog: one for elbow dysplasia and one for idiopathic epilepsy.

## Introduction

In genome-wide association studies (GWAS), the primary objective is generally to identify associations between a phenotype of interest and genetic variants. This involves assessing the p-values of certain test statistics by deriving the appropriate null distribution or an asymptotic approximation to it. However, this is not always feasible as the distribution may be intractable. This can arise when the test statistics result from black-box machine learning methods [1, 2], or in region-based tests where association signals over multiple sites are combined, e.g., rare variant tests [3], and/or when the test statistic involves data-adaptive weights [4]. A further limitation can arise when, even if the distribution (or the asymptotic distribution) of the test statistic is known for single tests, significance needs to be assessed for the maximum of many correlated tests. This occurs in genome scans to establish a genome-wide significance threshold, where the linkage disequilibrium present between the markers induces correlation between the association tests [5–7].

To overcome these limitations, a common approach is to perform permutation testing so as to obtain replicates of the data under the null hypothesis from which an empirical distribution can be derived [6, 7]. A fundamental assumption underlying permutation testing is exchangeability of the subjects in the sample, which is often approximately satisfied in population-based samples. However, this assumption can be violated in the presence of genetic structure in the sample (e.g., kinship, cryptic relatedness, and/or population structure), as it will introduce correlation in the sample, including in the phenotype values through polygenic effects [8, 9]. Hence, naive application of permutation testing will usually not preserve the correlation structure and can result in inflated type 1 error rates [10] (though this can be avoided if, for example, all subjects in the sample are equally related [11]). We consider here the problem of permutation testing for a binary trait in the presence of polygenic effects in a sample with population structure, cryptic and/or family relatedness. For quantitative traits, a permutation-based test that adjusts for the correlation structure has previously been proposed [9, 11]. The approach, called MVNpermute, is based on a linear mixed model (LMM) and is applicable for multivariate-normal data. It incorporates genetic relatedness through the inclusion of random effects in the LMM and can also adjust for covariates. However, as MVNpermute is primarily

designed for quantitative traits that are multivariate-normal, there will be more model misspecification when applied to binary traits as the binary nature of the data is not incorporated into the LMM.

We propose BRASS (for "binary trait resampling method adjusting for sample structure"), a permutation procedure for a binary trait which incorporates both covariates and the correlation structure present in the sample. In contrast to the LMM-based approach, it accommodates the binary nature of the trait through a quasi-likelihood framework that considers the effect of covariates on a logit scale in the mean structure as well as the relationship between the trait mean and its variance, both of which are important features of binary data (see, e.g., [12] for importance of modeling covariates in binary data). Hence, BRASS benefits from less model misspecification in the presence of important covariates relative to the LMM-based approach. Here, we show that the permutation-based replicates obtained with BRASS are able to maintain correct type 1 error control, and we compare it with several alternative approaches. We demonstrate the use of our method in a context of assessing genome-wide significance in domestic dog GWAS. In animal studies, permutation testing is commonly used when there is not a consensus threshold for genome-wide significance [13–16], and it is also used in some human studies when the consensus threshold might not be applicable [6]. We apply BRASS to perform association mapping in two GWAS of domestic dogs, one for elbow dysplasia (ED [OMIA: 000330–9615]) and and one for idiopathic epilepsy (IE [OMIA: 000344–9615]).

## Description of the methods

### Overview of BRASS method

We consider the problem of resampling a binary trait variable that is correlated across individuals, where the correlation can arise from, e.g., population structure or related individuals. Our aim is to derive replicates of the trait under the null hypothesis of no association while accounting for the correlation that is present in the sample but whose structure is unknown. To obtain these replicates, we start by modeling the response using a previously described quasi-likelihood framework for correlated binary traits [17, 18],

$$\boldsymbol{\mu} := \mathbb{E}(\mathbf{Y}|\mathbf{X}, \mathbf{G}) = \text{logit}^{-1}(\mathbf{X}\boldsymbol{\beta} + \mathbf{G}\gamma), \text{ and} \tag{1}$$

$$\boldsymbol{\Omega} := \text{Var}(\mathbf{Y}|\mathbf{X}, \mathbf{G}) = \boldsymbol{\Gamma}^{1/2} \boldsymbol{\Sigma} \boldsymbol{\Gamma}^{1/2}. \tag{2}$$

where $\mathbf{Y} = (Y_1,\ldots,Y_n)^T$ denotes the phenotype vector for $n$ subjects, $\mathbf{X}$ is the $n \times k$ matrix of $k$ covariates (including an intercept term), which is assumed to be of rank $k$, $\mathbf{G} = (G_1,\ldots,G_n)^T$ is the vector of genotypes at the marker of interest, where $G_i$ denotes the minor allele count for the $i$th individual, $\boldsymbol{\beta}$ is a $k$-length vector representing the unknown effects of covariates, $\gamma$ represents the unknown effect of the genetic marker, $\boldsymbol{\Gamma}$ is an $n \times n$ diagonal matrix with $i$th diagonal entry $\mu_i(1 - \mu_i)$ where $\mu_i$ is the $i$th element of $\boldsymbol{\mu}$, the $n \times n$ matrix $\Sigma$ is given by

$$\boldsymbol{\Sigma} = \xi \boldsymbol{\Phi} + (1 - \xi) \mathbf{I}_n, \tag{3}$$

where the unknown scalar parameter $\xi \in [0, 1]$ can be viewed as a heritability parameter, and the $n \times n$ matrix $\boldsymbol{\Phi}$ is generally a kinship or genetic relatedness matrix (GRM).

The framework characterized by (1) and (2) allows for modeling important covariates, such as biological or ancestry informative covariates, in the mean structure, while the residual correlation between subjects is captured directly within the variance structure by the inclusion of $\boldsymbol{\Phi}$ in (3). Furthermore, by including the matrix $\boldsymbol{\Gamma}$ in (2), this framework models the dependence of the phenotypic variance on the mean, a key feature of binary data.

Since we wish to simulate replicates of the binary trait under the null hypothesis of no association, $H_0 : \gamma = 0$, the remaining unknown parameters are $\boldsymbol{\beta}$ and $\xi$, which need to be estimated from the data. The null estimate $(\hat{\boldsymbol{\beta}}, \hat{\xi})$ is easily obtained by iteratively solving the following system of estimating equations [18] under the constraint $\gamma = 0$,

$$D_{\boldsymbol{\beta}}^T \boldsymbol{\Omega}^{-1}(\mathbf{Y} - \boldsymbol{\mu}) = \mathbf{0}, \tag{4}$$

$$(\mathbf{Y} - \boldsymbol{\mu})^T \boldsymbol{\Gamma}^{-1/2} \boldsymbol{\Sigma}^{-1}(\boldsymbol{\Phi} - \mathbf{I})\boldsymbol{\Sigma}^{-1}\boldsymbol{\Gamma}^{-1/2}(\mathbf{Y} - \boldsymbol{\mu}) = \text{trace}(\boldsymbol{\Sigma}^{-1}(\boldsymbol{\Phi} - \mathbf{I})), \tag{5}$$

where $D_{\boldsymbol{\beta}} = \frac{\partial \boldsymbol{\mu}}{\partial \boldsymbol{\beta}}$ is a Jacobian of the conditional mean of the trait with respect to $\boldsymbol{\beta}$. In our model with a logit link function, $D_{\boldsymbol{\beta}} = \boldsymbol{\Gamma} \mathbf{X}$.

The justification underlying an ordinary permutation test is an assumption of exchangeability under the null hypothesis, which does not generally hold in most situations of correlated data. Instead, we develop and apply an approximate permutation test based on second-order exchangeability, in which we permute a vector, $\hat{\boldsymbol{\zeta}}$, whose entries each have the same mean and variance and are uncorrelated with each other. Our approach builds on previous work [9, 11] that was limited to quantitative traits, and we extend the ideas to binary traits.

We start by fitting the null model given by Eqs 1 and 2 with $\gamma = 0$, and we let $\mathbf{Y} - \hat{\boldsymbol{\mu}}$ denote the resulting residual vector. We aim to map $\mathbf{Y} - \hat{\boldsymbol{\mu}}$ to a lower-dimensional orthonormal space in which its entries will be second-order exchangeable, with the result of this map being $\hat{\boldsymbol{\zeta}}$. To form $\hat{\boldsymbol{\zeta}}$, we need to first obtain the approximate covariance matrix of $\mathbf{Y} - \hat{\boldsymbol{\mu}}$ under the null hypothesis, which is challenging because $\mathbf{Y} - \hat{\boldsymbol{\mu}}$ has no closed-form expression as a function of the data.

To obtain the approximate covariance matrix, we first fix $\xi$ and let $U(\boldsymbol{\beta}) = D_{\boldsymbol{\beta}}^T \boldsymbol{\Omega}^{-1}(\mathbf{Y} - \boldsymbol{\mu})$, which corresponds to the quasi-score function for $\boldsymbol{\beta}$. Under the null hypothesis of $\gamma = 0$, let $\boldsymbol{\beta}_0$ denote the true value of $\boldsymbol{\beta}$, and let $\boldsymbol{\mu}_0, \boldsymbol{\Gamma}_0, \mathbf{D}_0$ and $\boldsymbol{\Omega}_0$ correspond to $\boldsymbol{\mu}, \boldsymbol{\Gamma}, D_{\boldsymbol{\beta}}$ and $\boldsymbol{\Omega}$, respectively, evaluated at $\boldsymbol{\beta}_0$ and $\gamma = 0$. Similarly, let $\hat{\boldsymbol{\mu}}, \hat{\boldsymbol{\Gamma}}, \hat{\mathbf{D}}$ and $\hat{\boldsymbol{\Omega}}$ be the same quantities evaluated at $\hat{\boldsymbol{\beta}}$ and $\gamma = 0$. We apply a Taylor series expansion to $U(\boldsymbol{\beta})$ around $\boldsymbol{\beta}_0$, evaluated at $\hat{\boldsymbol{\beta}}$:

$$U(\hat{\boldsymbol{\beta}}) \approx U(\boldsymbol{\beta}_0) + \frac{\partial U(\boldsymbol{\beta})}{\partial \boldsymbol{\beta}}\bigg|_{\boldsymbol{\beta}=\boldsymbol{\beta}_0} \cdot (\hat{\boldsymbol{\beta}} - \boldsymbol{\beta}_0). \tag{6}$$

Using the fact that $U(\hat{\boldsymbol{\beta}}) = \mathbf{0}$, replacing the Jacobian in (6) by its expectation (similar to Fisher scoring), and solving for $\hat{\boldsymbol{\beta}} - \boldsymbol{\beta}_0$ we get,

$$
\begin{aligned}
\hat{\boldsymbol{\beta}} - \boldsymbol{\beta}_0 \quad &\approx -\left\{ \mathbb{E}\left[ \frac{\partial U(\boldsymbol{\beta})}{\partial \boldsymbol{\beta}} \right]\bigg|_{\boldsymbol{\beta}=\boldsymbol{\beta}_0} \right\}^{-1} U(\boldsymbol{\beta}_0), \\
&= (\mathbf{D}_0^T \boldsymbol{\Omega}_0^{-1} \mathbf{D}_0)^{-1} \mathbf{D}_0^T \boldsymbol{\Omega}_0^{-1}(\mathbf{Y} - \boldsymbol{\mu}_0).
\end{aligned}
\tag{7}
$$

As $\hat{\boldsymbol{\mu}}$ is a non-linear function of $\hat{\boldsymbol{\beta}}$, a Taylor series expansion is performed on $\boldsymbol{\mu}$ around $\boldsymbol{\beta}_0$, and evaluated at $\hat{\boldsymbol{\beta}}$,

$$\hat{\boldsymbol{\mu}} \approx \boldsymbol{\mu}_0 + \mathbf{D}_0 (\hat{\boldsymbol{\beta}} - \boldsymbol{\beta}_0), \tag{8}$$

Combining (7) and (8), we get for the residual vector

$$
\begin{aligned}
\mathbf{Y} - \hat{\boldsymbol{\mu}} \quad &= \mathbf{Y} - \boldsymbol{\mu}_0 - (\hat{\boldsymbol{\mu}} - \boldsymbol{\mu}_0), \\
&\approx [\mathbf{I} - \mathbf{D}_0 \, (\mathbf{D}_0^T \, \boldsymbol{\Omega}_0^{-1} \, \mathbf{D}_0)^{-1} \mathbf{D}_0^T \, \boldsymbol{\Omega}_0^{-1}] \, (\mathbf{Y} - \boldsymbol{\mu}_0).
\end{aligned}
\tag{9}
$$

As a result, we can approximate the covariance matrix of the residuals as

$$
\mathrm{Var}(\mathbf{Y} - \hat{\boldsymbol{\mu}}) \approx \boldsymbol{\Omega}_0 - \boldsymbol{\Gamma}_0 \mathbf{X} \, (\mathbf{X}^T \boldsymbol{\Gamma}_0 \, \boldsymbol{\Omega}_0^{-1} \, \boldsymbol{\Gamma}_0 \mathbf{X})^{-1} \mathbf{X}^T \boldsymbol{\Gamma}_0.
\tag{10}
$$

The correlation present in $(\mathbf{Y} - \hat{\boldsymbol{\mu}})$ arises from two sources: that introduced by $\boldsymbol{\Phi}$ through $\boldsymbol{\Omega}_0$ and that introduced from using the estimated mean $\hat{\boldsymbol{\mu}}$ instead of the true unknown mean $\boldsymbol{\mu}_0$. We use a factorization $\mathbf{C}$ of $\boldsymbol{\Omega}_0$, with $\boldsymbol{\Omega}_0 = \mathbf{C}^T \mathbf{C}$, to remove the correlation due to $\boldsymbol{\Phi}$ and obtain,

$$
\mathrm{Var}[\mathbf{C}^{-T}(\mathbf{Y} - \hat{\boldsymbol{\mu}})] \approx \mathbf{I} - \mathbf{W}(\mathbf{W}^T \mathbf{W})^{-1} \mathbf{W}^T = \boldsymbol{\Psi}_0,
\tag{11}
$$

where $\mathbf{W} = \mathbf{C}^{-T} \boldsymbol{\Gamma}_0 \mathbf{X}$. The matrix $\boldsymbol{\Psi}_0$ in (11) is symmetric, idempotent and of rank $n - k$, so it can be expressed as $\boldsymbol{\Psi}_0 = \mathbf{V} \mathbf{V}^T$ where the columns of $\mathbf{V}$ contain the eigenvectors of $\boldsymbol{\Psi}_0$ corresponding to eigenvalue 1, and $\mathbf{V}^T \mathbf{V} = \mathbf{I}_{n-k}$. We use $\mathbf{V}^T$ to remove the remaining correlation that is driven by parameter estimation.

Applying these two linear transformations, we obtain $\boldsymbol{\zeta} = \mathbf{V}^T \mathbf{C}^{-T}(\mathbf{Y} - \hat{\boldsymbol{\mu}})$, which has covariance matrix

$$
\mathrm{Var}(\boldsymbol{\zeta}) \approx \mathbf{V}^T \boldsymbol{\Psi} \mathbf{V} = \mathbf{I}_{n-k}. \qquad (15)
\tag{12}
$$

Thus, we obtain a transformation of the residuals where the entries have approximately the same variance and are uncorrelated.

In practice $\xi$ is also unknown, and the matrices $V$ and $C$ are functions of both the unknown parameters $\xi$ and $\boldsymbol{\beta}_0$, so we plug in the values $\hat{\xi}$ and $\hat{\boldsymbol{\beta}}$ estimated under the null hypothesis and call the resulting matrices $\hat{V}$ and $\hat{C}$. Finally we obtain $\hat{\boldsymbol{\zeta}} = \hat{V}^T \hat{C}^{-T}(Y - \hat{\boldsymbol{\mu}})$. For each permutation replicate, we draw a random permutation matrix $\boldsymbol{\Pi}$ and obtain $\boldsymbol{\Pi}\hat{\boldsymbol{\zeta}}$, a permuted version of $\hat{\boldsymbol{\zeta}}$, which we then back-transform to the residual space by applying $\hat{C}^T \hat{V}$. Finally, we add on $\hat{\boldsymbol{\mu}}$ in order to obtain the trait replicate:

$$
\mathbf{Y}_\pi = \hat{\boldsymbol{\mu}} + \hat{\mathbf{C}}^T \hat{\mathbf{V}} \boldsymbol{\Pi} \hat{\mathbf{V}}^T \hat{\mathbf{C}}^{-T}(\mathbf{Y} - \hat{\boldsymbol{\mu}}).
\tag{13}
$$

Under the identity permutation, meaning that $\boldsymbol{\Pi} = \mathbf{I}$, we would recover the original response vector. We note that because $\hat{\boldsymbol{\Gamma}}$ is just a diagonal matrix, $\hat{\mathbf{C}}$ can be obtained from a factorization (e.g., cholesky or eigen decomposition) of $\hat{\boldsymbol{\Sigma}} = \hat{\xi} \, \boldsymbol{\Phi} + (1 - \hat{\xi}) \, \mathbf{I}$, which in turn could be obtained from a factorization of $\boldsymbol{\Phi}$ if desired, as was done in [9].

Examining the form of (13), there are three main steps used to obtain a transformation of the residuals with second-order exchangeable entries. The first is to center the response by using the estimated phenotypic mean. The second is to remove the correlation present due to polygenic effects, which is done by the pre-multiplication by $\hat{\mathbf{C}}^{-T}$. The last step is to remove the correlation that is generated from using parameter estimates instead of the true values when centering, and is represented by the pre-multiplication by $\hat{\mathbf{V}}^T$. It is possible that the last step might have only a minor impact on the method for large sample sizes and low number of parameters, though we did not investigate this. The replicates generated from this approach are quantitative, i.e., the binary nature of the response is not preserved during resampling. In subsection **Additional resampling methods considered**, we describe how the replicates can be transformed to be binary, and we consider both approaches in simulations. The fixed effects

and covariance structure estimated under the null are preserved in the trait replicates, making our method applicable to general scenarios where there are important covariates and/or population structure.

## LogMM-PQL

A logistic mixed model (LogMM) is a natural modeling choice for correlated binary data, and an alternative approach to BRASS for generating replicates of correlated binary data could be to fit a LogMM under the null hypothesis and then generate binary replicates from the fitted model. We have developed such an approach, which we call LogMM-PQL. The model underlying LogMM-PQL is that, conditional on $\mathbf{X}$, $\mathbf{G}$ and $\mathbf{u}$, $Y_i$ is Bernoulli($m_i$), independently across $i = 1, \ldots, n$, where $m_i$ is the $i$th component of the vector

$$\mathbf{m} := E(\mathbf{Y}|\mathbf{X}, \mathbf{G}) = \mathrm{logit}^{-1}(\mathbf{X}\boldsymbol{\beta} + \mathbf{G}\gamma + \mathbf{u}), \tag{14}$$

where $\boldsymbol{\Phi}$, $\mathbf{Y}$, $\mathbf{X}$, $\mathbf{G}$, $\boldsymbol{\beta}$ and $\gamma$ are as before; $\mathbf{u}$ is a multivariate normal random effects vector of length $n$ with $\mathbf{u} \sim \mathrm{MVN}(0, \sigma^2\boldsymbol{\Phi})$ where $\sigma^2$ is an unknown scalar, and where $\gamma$ will be set to zero because the model will be fitted under the null hypothesis of no effect.

The major difficulty with use of LogMMs in the GWAS context is that they are computationally challenging to fit as they involve high-dimensional integrals. Therefore, their use can involve a trade-off between computational feasibility and accuracy of the results. For the parameter estimation part of LogMM-PQL, we use a fast penalized quasi-likelihood (PQL) approach, as implemented in GMMAT [19]. With $\gamma$ set to zero, we use GMMAT to obtain estimates $\hat{\boldsymbol{\beta}}$ and $\hat{\sigma}^2$, and then generate trait replicates by drawing random vectors $\mathbf{Y}$ from the LogMM specified by Eq (14) with $(\boldsymbol{\beta}, \gamma, \sigma^2)$ set equal to $(\hat{\boldsymbol{\beta}}, 0, \hat{\sigma}^2)$. LogMM-PQL renders binary replicates that incorporate the mean and variance structure specified by the fitted logistic model.

## Additional resampling methods considered

In addition to BRASS and LogMM-PQL, the other resampling methods we compare in simulations include the previously proposed MVNpermute [9, 11], as well as a method we call Naive, which involves fitting a LMM whose form under the null is $\mathbf{Y} = \mathbf{X}\boldsymbol{\beta} + \mathbf{e}$, with $\mathbf{e} \sim \mathrm{MVN}(0, \sigma_1^2\boldsymbol{\Phi} + \sigma_2^2\mathbf{I})$. After obtaining the estimated coefficients $\hat{\boldsymbol{\beta}}$, the Naive method replicates are obtained by permuting the residuals $(\mathbf{Y} - \mathbf{X}\hat{\boldsymbol{\beta}})$, and adding back $\mathbf{X}\hat{\boldsymbol{\beta}}$.

While the LogMM-PQL replicates are binary, those from BRASS, MVNpermute and Naive are not. However, we consider alternative approaches based on converting these replicates to binary, referred to as BRASS$_{mod}$, MVNpermute$_{mod}$ and Naive$_{mod}$, respectively, where the subscript "mod" indicates that the trait replicate has been modified to convert it to binary. To convert a continuous trait replicate $\mathbf{Y}_\pi$ to binary, a threshold is set and the values of $\mathbf{Y}_\pi$ above and below that threshold are converted to 1 and 0, respectively, with the threshold chosen so that the original trait $\mathbf{Y}$ and the binary replicate $\mathbf{Y}_\pi$ have the same case/control proportions.

The features of the various methods are summarized and compared in S1 Table.

## Modeling sample structure in the replicates

All 7 resampling strategies described above involve fitting a trait model that incorporates the structure present in the sample. A common approach to modeling trait correlation due to additive polygenic effects is to include a GRM in a LMM for the trait so as to model the structure using random effects [20]. Alternatively, the top principal components of a given GRM can be used as covariates to model the structure present using fixed effects [21]. Here, we allow

either of the two approaches or they can be combined (i.e., both fixed and random effects can be used) to capture the sample structure in the null model. To combine the two approaches, we use the top PCs from a GRM as fixed effects in the null model and then build a new GRM that is adjusted to exclude the effects of these top PCs in order to capture the leftover structure as random effects. In the simulations and data analysis, we form the new GRM using equation 4 of a previous work [22].

When generating replicates, it is of course important to accurately model the structure present in the sample, but in addition, it is also important to avoid creating new structure (not present in the data) as an artifact of the replication procedure. To do this, we recommend the use of leave-one-chromosome-out (LOCO) GRMs [23, 24] in the replication procedure. In that case, replicates would be generated separately for each chromosome, with the SNPs for the given chromosome left out of the GRM. This avoids introducing new trait-SNP association in the replicates.

## Methods for simulation studies

We perform simulation studies to compare the effectiveness of the 7 proposed permutation-based methods for generating binary trait replicates in the context of correcting for multiple testing, with both population structure and pedigree structure present in the sample. We consider a setting in which a binary trait is tested for association with each of $m$ markers and the aim is to assess the significance of the smallest p-value out of the $m$ association tests. This context would arise in testing significance of a genomic region or in assessing genome-wide significance. We simulate non-causal markers that are tested for association with a binary trait and estimate the significance threshold for the top signal amongst them using the empirical distribution from trait replicates generated under the null hypothesis of no association. Genotype, trait and covariates are generated under multiple simulation settings. In real data applications, the true trait model is usually not known a priori, and it can be important to assess the impact of model misspecification on type 1 error [25]. We consider the effects of model misspecification due to (1) assuming a logistic link function when the true model is a liability threshold model and (2) exclusion of an important covariate from the model. In addition, we investigate the impact of ascertainment in the sample as case-control studies often involve phenotype-based ascertainment which can introduce misspecification in the model as well.

For each setting, we perform a simulation study consisting of 4 stages: (1) data set creation, i.e., simulation of genotype and trait data under the null hypothesis of no association, with 20,000 data sets per setting; (2) generation of permutation replicates, in which we consider every possible pair $(i, j)$ of simulated data set $i$ and permutation method $j$, for $1 \leq i \leq 20,000$ and $1 \leq j \leq 7$, and for each of these, 10,000 permutation replicates are drawn for data set $i$ using permutation method $j$; (3) analysis of permutation replicates, where for each trio $(i, j, k)$ of simulated data set $i$, permutation method $j$, and permutation replicate $k$, $1 \leq k \leq 10,000$, the trait in permutation replicate $k$ is tested for association with each of the $m$ non-causal markers in data set $i$, the smallest p-value is recorded in each case, and then the resulting 10,000 p-values for each pair $(i, j)$ are used to estimate a genomewide significance threshold (at level $\alpha = 10^{-3}$) for data set $i$ based on permutation method $j$; and (4) evaluation of the type 1 error rate of the genomewide threshold for each permutation method $j$ by comparing the smallest p-value for each data set $i$ to the threshold determined in (3) for pair $(i, j)$ and assessing whether the overall proportion of rejections among the 20,000 data sets is significantly different from $\alpha$ for method $j$. We now describe each of these stages.

For stage (1) data set creation, we simulate genotypes using a 2 sub-population Balding-Nichols model [26] with $F = .01$ and ancestral allele frequencies for SNPs drawn independently

and uniformly on (.2, .8). We simulate multiple pedigrees of the same configuration (S1 Fig), with equal numbers of pedigrees assigned to each of the two sub-populations. Founder alleles for a pedigree are assumed to be drawn from the sub-population to which the pedigree is assigned, and gene dropping is used to determine genotypes for other pedigree members. We simulate phenotypes according to two different generative models for the trait. In each case, $M_c = 10^4$ causal SNPs are simulated as well as 3 covariates: age, sex and an i.i.d. standard normal covariate. The first generative model we use is logistic:

$$Y_i | \mathbf{X}_i, \mathbf{W}_i, \boldsymbol{\alpha} \sim \text{Bernoulli}(p_i), \text{ independently, with } \text{logit}(p_i) = \mathbf{X}_i \boldsymbol{\beta} + \mathbf{W}_i \boldsymbol{\alpha}, \qquad (15)$$

where $\mathbf{X}_i$ is the covariate row vector for the $i$th individual; $\boldsymbol{\beta}$ are the fixed effects for the covariates; $\mathbf{W}_i$ is a row vector representing the SNP information for the $i$th individual corresponding to the $M_c$ causal markers standardized to have mean 0 and variance 1; $\boldsymbol{\alpha} = (\alpha_1, \cdots, \alpha_{M_c})^T \sim MVN(0, \sigma^2 \mathbf{I})$ represent the random effects of the $M_c$ causal markers with resulting total additive polygenic variance $\sigma_a^2 = \sigma^2 M_c$. The values for $\boldsymbol{\beta}$ and $\sigma_a^2$ are chosen so that, considering the variance of $\text{logit}(p_i)$ due to covariate effects and additive polygenic random effects, the fraction of this variance due to covariates is fixed at 20%, 40%, 60% or 80% (while the remaining fraction is due to additive polygenic random effects), and so that Bernoulli error explains on average either (a) about 20% of the total phenotypic variability, resulting in a prevalence of 30% or (b) about 55% of the total phenotypic variability, resulting in a prevalence of 5%. In addition, the effect size of the standard normal covariate is set so that the p-value for its significance in a LMM Wald test is on average 0.05. (This is the covariate that is set to be missing in the settings in which an important covariate is missing from the observed data set.) The second generative model we use is a liability threshold model:

$$Y_i = \mathbb{1}_{\{L_i > 0\}}, \text{ with } L_i = \mathbf{X}_i \boldsymbol{\beta} + \mathbf{W}_i \boldsymbol{\alpha} + \epsilon_i, \qquad (16)$$

where $L_i$ is the latent liability for individual $i$, and $\boldsymbol{\epsilon} = (\epsilon_1, \ldots, \epsilon_n)^T \sim MVN(0, \sigma_e^2 \mathbf{I})$. The values of $\boldsymbol{\beta}$, $\sigma_a^2$, and $\sigma_e^2$ are chosen using analogous constraints as for the logistic model, where the phenotypic variability due to $\boldsymbol{\epsilon}$ is constrained to be about 20% of the total phenotypic variability, and the prevalence is set to either 30% or 5%. In both 15 and 16, larger values of $\sigma_a^2$ correspond to more severe confounding effects of population/family structure on phenotype-genotype association. For the settings in which ascertainment is used, we simulate data under either the model in (15) or that in (16) and select 1,000 individuals at random to be retained in the sample using either (a) a balanced case-control ratio or (b) a 3:7 case-control ratio. In all settings, in addition to the $M_c$ causal SNPs, $m = 100$ non-causal SNPs are generated as above and used for testing under the null hypothesis of no association. (Further details on the stage (1) simulations can be found in S1 Text.)

We now describe stage (2) generation of permutation replicates for each data set and for each of the 7 permutation methods (BRASS, LogMM-PQL, MVNpermute, Naive, BRASS$_{mod}$, MVNpermute$_{mod}$, and Naive$_{mod}$). For a given data set, the same covariate and GRM information is input to each of the 7 permutation methods. The GRM $\boldsymbol{\Phi}$ is calculated using the $M_c$ causal markers, and then $D = 1$ top PC is removed from $\boldsymbol{\Phi}$ using equation 4 of a previous work [22], resulting in adjusted GRM $\tilde{\boldsymbol{\Phi}}$, which is the GRM input to each permutation method. In most settings, the 4 covariates input to each permutation method are the top PC plus the 3 covariates used to generate the trait. In the settings in which an important covariate is assumed to be missing from the data set, the 3 covariates input to each permutation method are the top PC, age and sex, with the normal covariate left out.

In stage (3), for each pair $(i, j)$ of data set and permutation method, the $10^4$ permutation replicates from method $j$ are each tested for association against the panel of $m$ SNPs in data set

$i$, and the smallest p-value among the $m$ SNPs is recorded for each permutation replicate, resulting in $10^4$ p-values for pair $(i, j)$. The genome-wide threshold value, corresponding to significance level $\alpha$, for pair $(i, j)$ is estimated by the $100\alpha$th percentile of the p-values observed for that pair. In principle, any association testing method could be used for the stage (3) association tests, provided that the same method is also used in stage (4). One consideration is that only four of the permutation methods we examine create binary replicates, while the other three create non-binary, real-valued replicates. Whether having non-binary replicates is problematic or not depends on the method of analysis. For example, it is not a problem if the data are analyzed using a linear mixed model or using the CARAT [18] or CERAMIC [17] binary trait methods, because these methods are based on solving certain estimating equations and work well provided that the first and second moments of the data are appropriately modeled. However, analysis by ordinary logistic regression would require binary data. To be able to test all 7 methods, we therefore use a (slightly modified) version of CARAT for the stage (3) and (4) association tests, where the modification is in the estimation of $\sigma_g^2$, the genotypic variance parameter for CARAT. In the original CARAT paper [18] $\sigma_g^2$ is estimated by $\hat{\sigma}_g^2 = 2\hat{f}(1 - \hat{f})$ with $\hat{f} = .5\bar{G}$, where $\bar{G}$ is the sample average of $\boldsymbol{G}$ in the data. Our modification is to replace this with $\tilde{\sigma}_g^2 = \mathbf{G}^T \mathbf{P} \mathbf{G}/(n - D - 1)$ where $D$ is the number of PCs removed from the GRM, and $\mathbf{P} = \tilde{\boldsymbol{\Phi}}^{-1} - \tilde{\boldsymbol{\Phi}}^{-1}\mathbf{Z}(\mathbf{Z}^T\tilde{\boldsymbol{\Phi}}^{-1}\mathbf{Z})^{-1}\mathbf{Z}^T\tilde{\boldsymbol{\Phi}}^{-1}$, where $\mathbf{Z}$ is the $n \times (D + 1)$ matrix whose columns are the $D$ eigenvectors that are removed as well as a columns of ones. (The rationale for this modification is detailed in S1 Text.) When we apply CARAT, we use $\tilde{\boldsymbol{\Phi}}$ as the GRM, and we include as fixed effects the top PC plus the 3 covariates that were used to generate the trait. For the settings when an important covariate is assumed to be missing from the data set, the 3 covariates used in the analysis of replicates are the top PC, age and sex, with the normal covariate left out.

Finally, in stage (4), we evaluate the type 1 error of the genomewide threshold for each permutation method $j$ by by comparing the p-value for each data set $i$ to the threshold determined in stage (3) for the pair $(i, j)$ and assessing whether the overall proportion of rejections out of 20,000 data sets is significantly different from the nominal level $\alpha$ for method $j$. To make our massive simulation project more computationally efficient, we actually used an adaptive procedure so that, e.g., if a dataset is sufficiently far from the significance threshold based on analysis of the first $10^3$ permutation replicates, it would not be necessary to analyze the remainder of the $10^4$ permutation replicates (see S1 Text for details). The amount of downward bias introduced in the empirical type 1 error estimate by this procedure is very small compared to the Monte Carlo sampling variability. For example, when level .01 is tested under the null hypothesis with 20,000 Monte Carlo replicates using the adaptive procedure, the squared bias in the type 1 error estimate due to the adaptive procedure is less than 5e-10, which is several orders of magnitude smaller than the Monte Carlo error variance of 5e-07.

## Verification and comparison

### Assessment of type 1 error

We assess and compare the performance of the 7 proposed resampling methods in the context of correcting for multiple testing, with both population and pedigree structure present in the sample in addition to important covariates. Fig 1 shows the empirical type 1 error rates of all methods, compared to the nominal, when the true trait model is logistic. The type 1 error of BRASS is well-controlled in all settings. For LogMM-PQL, we observe significant inflation in the type 1 error rate, with the amount of inflation increasing as the confounding due to structure present in the sample increases. This likely occurs because the PQL model fitting

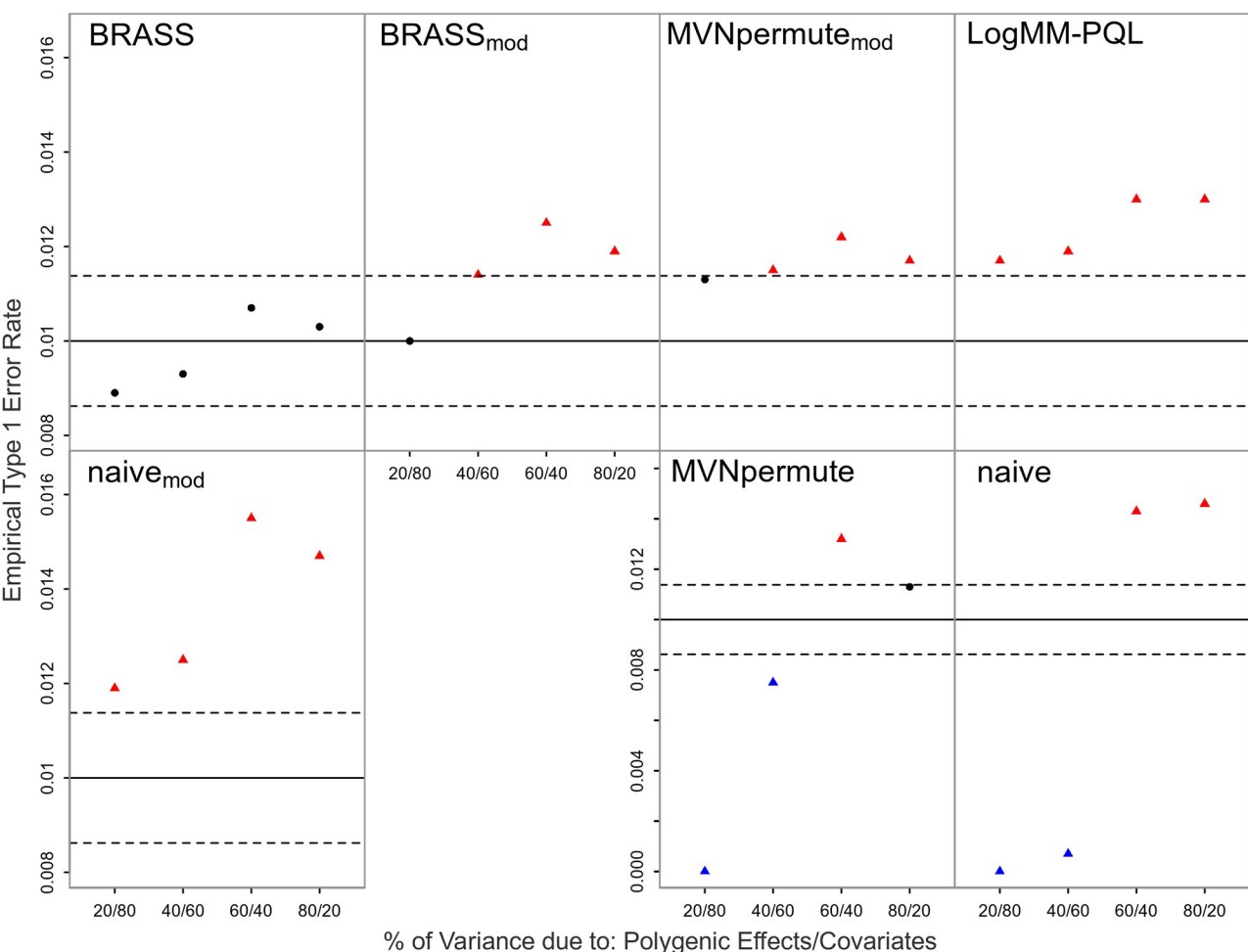

**Fig 1. Empirical Type 1 Error Rates with Logistic Model Including all Covariates at Nominal Level 0.01.** The error rate is based on 20,000 simulated replicates. The solid horizontal line represents the nominal level and the dashed lines represent rejection bounds outside of which the z-test comparing the estimated type 1 error to the nominal level is rejected at level .05. Estimates inside the rejections bounds are represented by circles and those outside the bounds are represented by triangles. Red, black and blue symbols represent inflated, well-controlled and conservative type 1 error rates, respectively. The proportion of variability on the logit scale attributable to polygenic effects vs. covariates is varied from 20 to 80% in increments of 20%. Prevalence is 30%.

approach tends to underestimate the variance component and hence, results in less correlation present in the simulated data compared to that in the original data. We observe that permuting the residuals from a linear mixed model (Naive) leads in all settings to lack of control of the type 1 error. More precisely, when covariates explain most of the variation on the logit scale, the method is overly conservative and when instead polygenic effects explain most of the variability, the type 1 error rate is significantly inflated. Moreover, we also find that adjusting for sample correlation prior to permutation based on LMM residuals, as done in MVNpermute, still leads to a very conservative test when covariates highly influence the variability on the logit scale. This is most likely because a linear mixed model does not allow any dependence between the trait variance and the mean, which is influenced by the covariates. As covariates become less important, we do see somewhat better control of the type 1 error. For Naive$_{mod}$ and MVNpermute$_{mod}$, control of the type 1 error is improved compared to the corresponding original methods when covariates strongly influence the phenotypic mean, and for BRASS$_{mod}$,

type 1 error control is worse compared to BRASS. All of the methods except BRASS show inflated type 1 error in at least one setting, whereas BRASS maintains the correct type 1 error rate in all settings.

## Robustness to model misspecification: Liability threshold vs. logistic model

As the true model for the trait is generally unknown in an association study, we are interested in assessing robustness of the type 1 error results to model misspecification. Thus, we also perform simulations in which the phenotype is simulated from a liability threshold model, which is more dissimilar to the models fit by BRASS and LogMM-PQL than logistic is. The type 1 error results for the liability threshold model are shown in Fig 2. The primary effect of the model misspecification in this case seems to be to make all the methods except Naive slightly more conservative.

## Robustness to missing an important covariate

We assess the robustness of the type 1 error results when an important covariate is omitted from both the fitted model and the model used for generating replicates. As it is usually not known a priori which variables should be kept in the analysis, one would commonly try including different combinations of covariates to finally determine the ones to include in the final model. It may occur that one of the covariates has a moderate effect on the trait and leads to a p-value close to the significance threshold (e.g. 0.05). Hence, a judgment call would be required for whether to keep the covariate in the model and one may decide to exclude it. It is thus of interest to see how the proposed methods would fare in such a scenario as the replicates generated would come from a more misspecified model.

The simulation results are displayed in S2 Fig. Similarly to the previous model misspecification results, the general effect is to make the results more conservative for all permutation methods except Naive.

## Robustness to ascertainment

We determine the robustness of the proposed methods when trait-based ascertainment has been applied to the sample. This is commonly used in case-control studies where individuals are included in the sample based on their disease affection status, such as if the prevalence of the trait in the population is too low to obtain sufficient power.

The results are shown in Figs 3 and 4, where it can be seen that BRASS is the only method that retains good control of the type 1 error rate in all of the ascertainment settings considered. In contrast, both logMM-PQL and Naive$_{mod}$ show significantly inflated type 1 error in every setting, while both MVNpermute and Naive give excessively conservative results when covariates have a major impact on the trait and tend to give significantly inflated results when the effect of covariates is low.

## Robustness to ascertainment, model misspecification and missing covariate, combined

Finally, we consider a scenario that simultaneously combines multiple features of the previous settings: (1) ascertainment from a prevalence of 5% to a case-control ratio of 3:7; (2) model misspecification in which the true model is a liability-threshold model, which is more dissimilar to the models fit by BRASS and LogMM-PQL than logistic is; and (3) model misspecification due to an important covariate being missing. The results are shown in Fig 5. As before, the model misspecification due to the liability threshold model and the omission of an

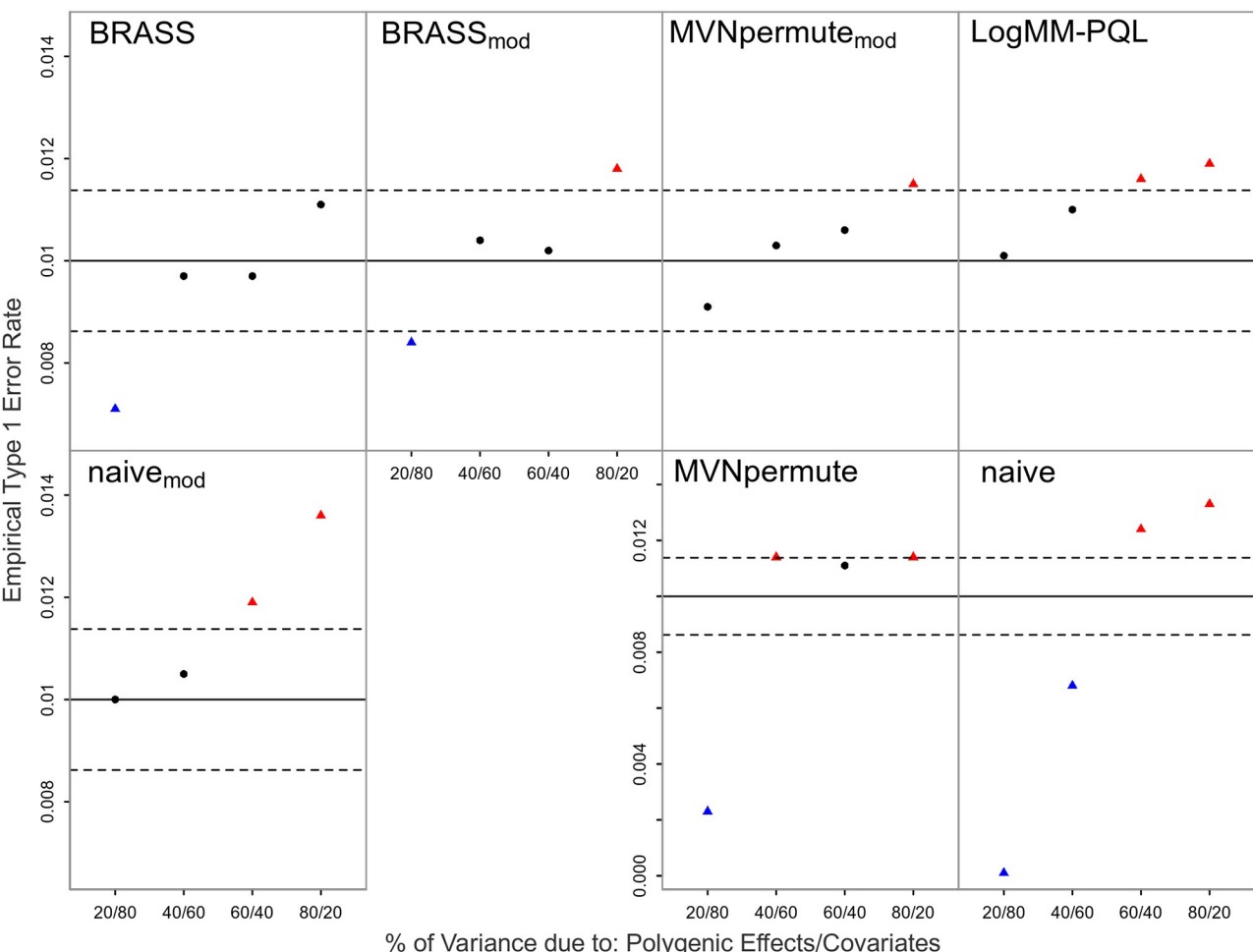

**Fig 2. Empirical Type 1 Error Rates with Liability Threshold Model at Nominal Level 0.01.** The model is misspecified with the true model being a liability threshold model. The error rate is based on 20,000 simulated replicates. The solid horizontal line represents the nominal level and the dashed lines represent rejection bounds outside of which the z-test comparing the estimated type 1 error to the nominal level is rejected at level .05. Estimates inside the rejections bounds are represented by circles and those outside the bounds are represented by triangles. Red, black and blue symbols represent inflated, well-controlled and conservative type 1 error rates, respectively. The proportion of variability on the liability scale attributable to polygenic effects vs. covariates is varied from 20 to 80% in increments of 20%. Prevalence is 30%.

important covariate have the effect of making most of the methods more conservative, where this effect seems most pronounced when covariate effects are much more important than polygenic effects (left-most case corresponding to "20/80" in each plot).

From the results across all the simulation settings, with and without ascertainment or model misspecification, it can be seen that BRASS provides the best type 1 error control and is the only method that does not have significantly inflated type 1 error in any setting.

## Applications

We apply BRASS to the problem of determining genome-wide significance for GWAS of two different traits in domestic dog. Unlike with humans, a genome-wide threshold for significance in domestic dog GWAS has not been well established. We analyze data from a large domestic dog study [27], with 4,224 dogs representing over 150 breeds genotyped at 185,805

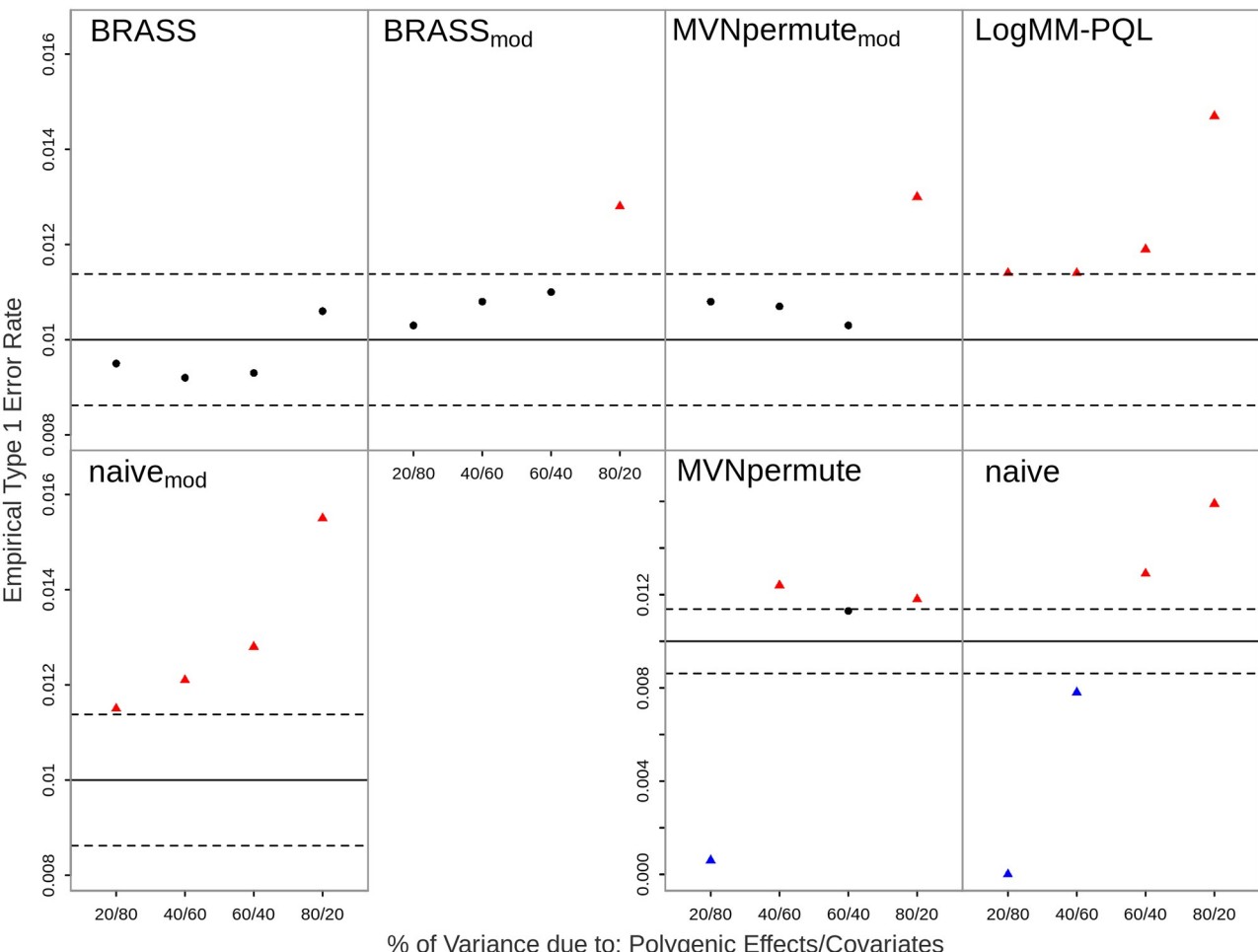

**Fig 3. Empirical Type 1 Error Rates with Ascertainment Scenario 1 at Nominal Level 0.01.** The error rate is based on 20,000 simulated replicates. The solid horizontal line represents the nominal level and the dashed lines represent rejection bounds outside of which the z-test comparing the estimated type 1 error to the nominal level is rejected at level .05. Estimates inside the rejections bounds are represented by circles and those outside the bounds are represented by triangles. Red, black and blue symbols represent inflated, well-controlled and conservative type 1 error rates, respectively. The proportion of variability on the logit scale attributable to polygenic effects vs. covariates is varied from 20 to 80% in increments of 20%. Prevalence is 30%, and ascertainment results in a case-control ratio of 1:1.

SNPs with multiple phenotypes recorded. We analyze two binary traits from this study, ED, for which the data contain 113 cases and 633 controls among 82 breeds, and IE, for which there are 34 cases and 168 controls from the Irish Wolfhound breed. For each trait, we compute a GRM after LD pruning and filtering for MAF < .05. For ED, we take the top 10 PCs out of the GRM and include them as covariates, and sex is included as a covariate for both traits. The data are then analyzed with CARAT. To generate trait replicates using BRASS, we create LOCO GRMs for each chromosome, where for ED we also take out the top 10 PCs in each case and include them as covariates, and sex is included as a covariate for both traits. For genome-wide significance assessment for each trait, we used BRASS to generate 10,000 replicates under the null hypothesis of no association. We analyzed each replicate in the same way that we analyzed the data and used the results of these analyses to form an empirical distribution of the test statistic under the null hypothesis. Genome-wide p-values were then estimated using the empirical distribution for each trait.

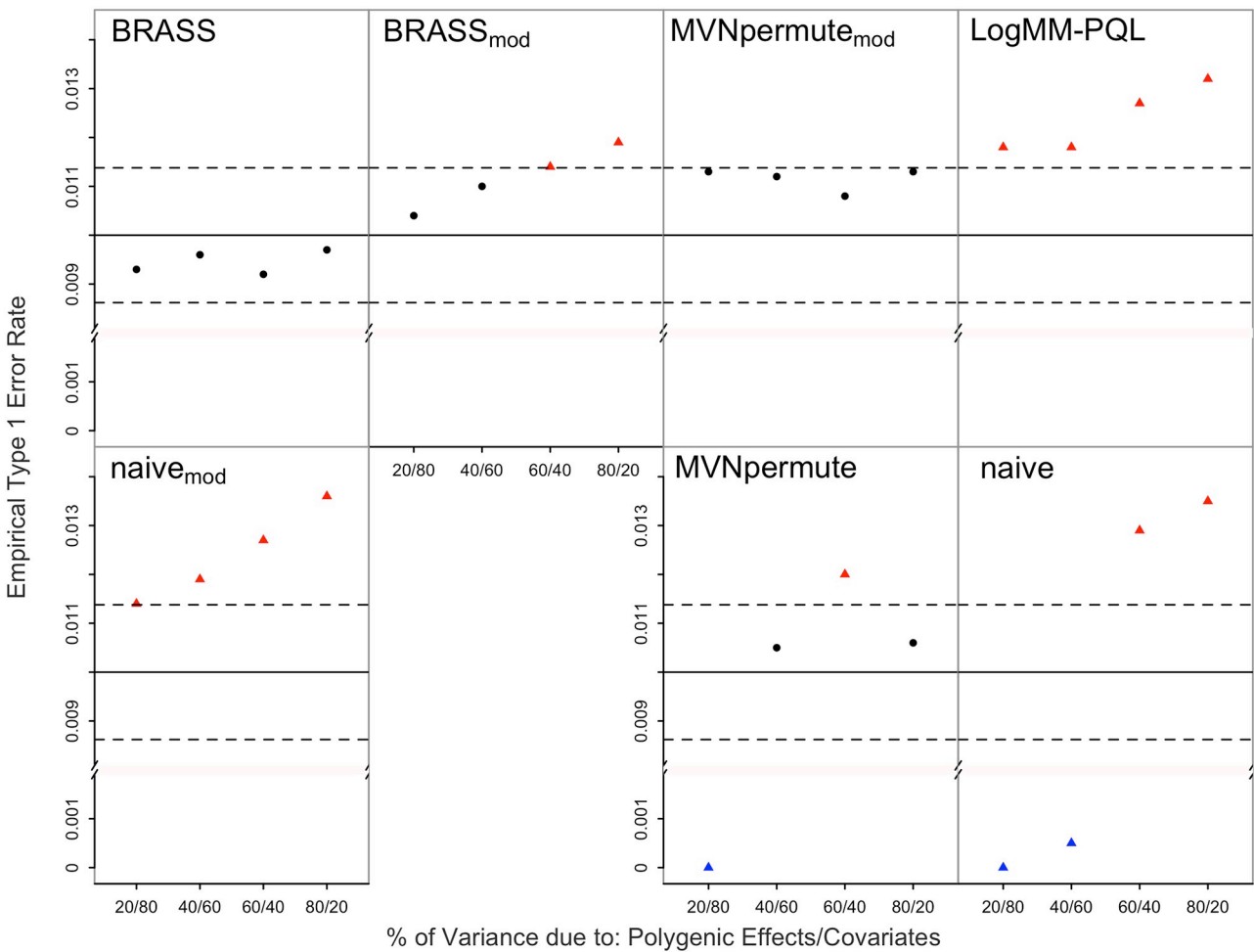

**Fig 4. Empirical Type 1 Error Rates with Ascertainment Scenario 2 at Nominal Level 0.01.** The error rate is based on 20,000 simulated replicates. The solid horizontal line represents the nominal level and the dashed lines represent rejection bounds outside of which the z-test comparing the estimated type 1 error to the nominal level is rejected at level .05. Estimates inside the rejections bounds are represented by circles and those outside the bounds are represented by triangles. Red, black and blue symbols represent inflated, well-controlled and conservative type 1 error rates, respectively. The proportion of variability on the logit scale attributable to polygenic effects vs. covariates is varied from 20 to 80% in increments of 20%. Prevalence is 5%, and ascertainment results in a case-control ratio of 3:7.

Manhattan plots of the p-values of the single-SNP tests for the observed data are presented in Fig 6 for both ED and IE phenotypes, along with the estimated genome-wide significance thresholds, which are estimated at nominal level 0.05 from the empirical distribution of the top association signal based on the trait replicates. The genomic control inflation factors for ED and IE are 1.02 and 0.98, respectively. For IE, we detect a previously-identified [27] 12Mb region on chromosome 4 (position 7.5–19.3 Mb) that reaches the BRASS genome-wide significance threshold, with nominal p-value $2.1 \times 10^{-8}$ corresponding to a genomewide p-value of .0008 obtained from BRASS. For ED, we detect 3 loci, summarized in Table 1. Compared to a previous reference [27], we detected one additional significant locus (rs23910667 on chromosome 35) that is not detected there. The genomewide p-values by the other 6 methods for these same SNPs (detailed in S1 Text) are somewhat smaller than those from BRASS, which is in line with the simulation studies showing that the other methods tend to be anti-conservative

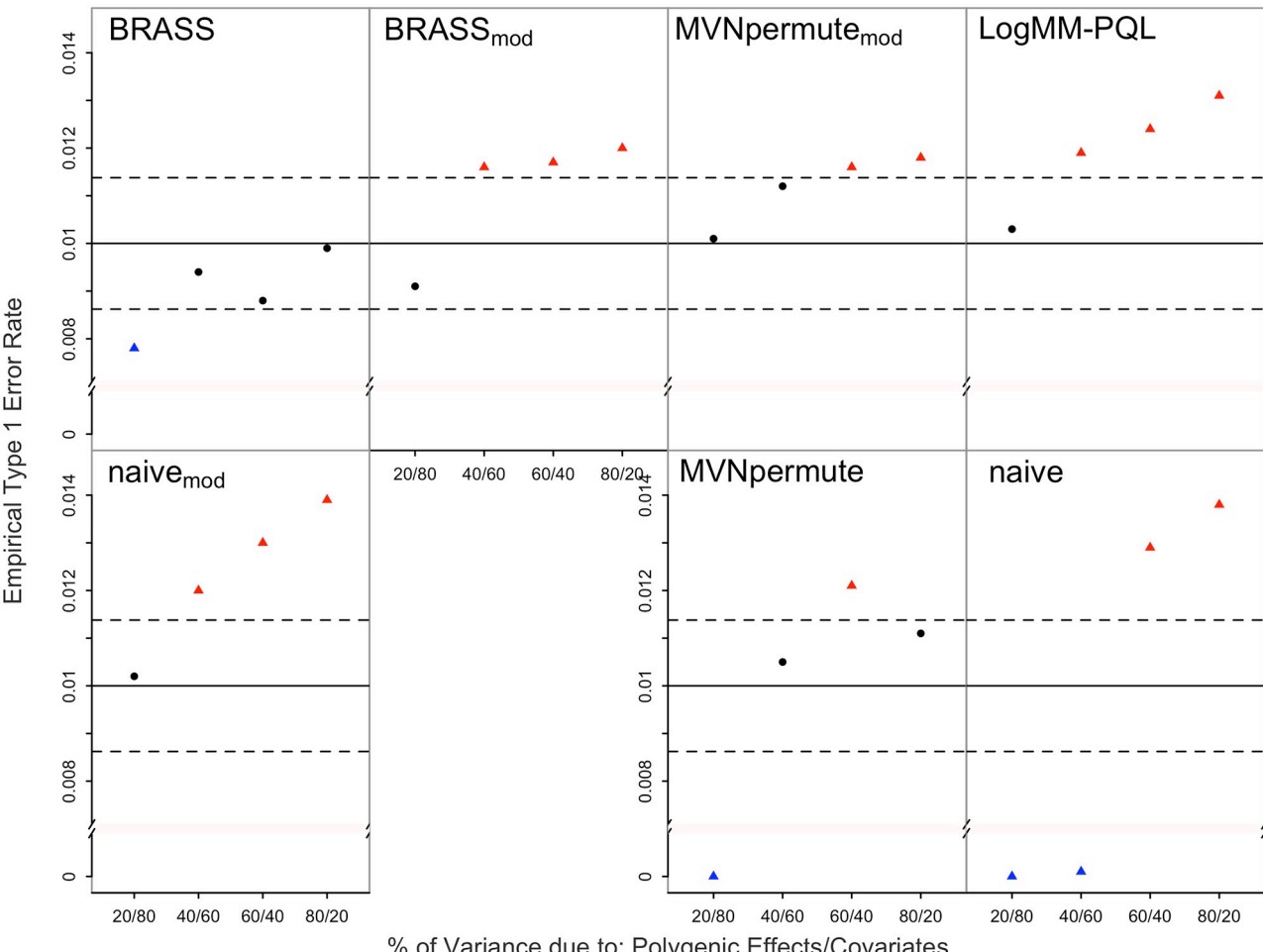

**Fig 5. Empirical Type 1 Error Rates with Ascertainment, ModelMisspecification and Missing Covariate at Nominal Level 0.01.** The model is misspecified with the true model being a liability threshold model. The error rate is based on 20,000 simulated replicates. The solid horizontal line represents the nominal level and the dashed lines represent rejection bounds outside of which the z-test comparing the estimated type 1 error to the nominal level is rejected at level .05. Estimates inside the rejections bounds are represented by circles and those outside the bounds are represented by triangles. Red, black and blue symbols represent inflated, well-controlled and conservative type 1 error rates, respectively. The proportion of variability on the liability scale attributable to polygenic effects vs. covariates is varied from 20 to 80% in increments of 20%. Prevalence is 5%, and ascertainment results in a case-control ratio of 3:7. The effect of the omitted covariate on the trait correspond to a Wald test p-value of .05 using a linear mixed model.

overall and in particular for models in which polygenic effects are more important than covariates.

## Computation time

In order to generate replicates from BRASS, the null model given by Eqs 1 and 2 with $\gamma = 0$ needs to be fitted, which includes specifying covariates and a GRM $\boldsymbol{\Phi}$. If not provided, obtaining the eigen decomposition of $\boldsymbol{\Phi}$ is the main computationally intensive step involved in fitting the null model [18]. However, this step (or one of equivalent computational complexity) is also typically performed when analyzing the data or when using permutation methods based on a linear or logistic mixed model. Once parameter null estimates are obtained, the transformed residuals $\hat{\zeta}$ as well as the linear map $\hat{\mathbf{C}}^T\hat{\mathbf{V}}$ in (13) need to be computed; however, once obtained, they can be re-used when generating trait replicates. As the eigen decomposition of

## A. ED

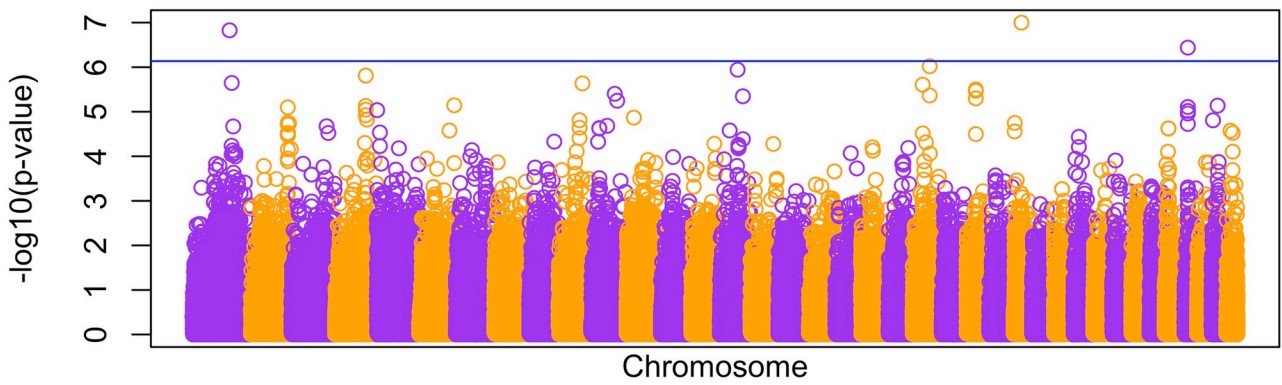

## B. IE

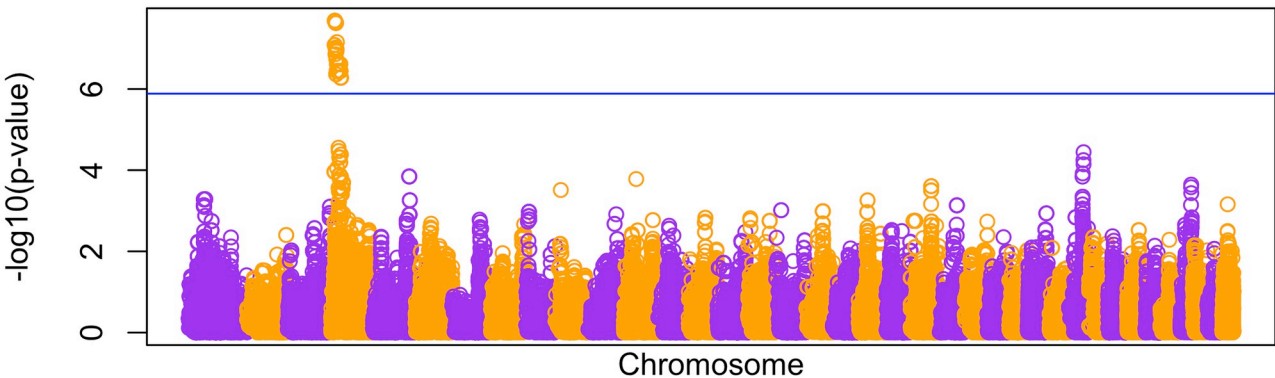

**Fig 6. Genome screen results in the domestic dog data.** Manhattan plots of the single-SNP association p-values using CARAT (on—$\log_{10}$ scale) for (A) ED and (B) IE, with genomic position on the *x*-axis. The horizontal lines represent estimated genome-wide thresholds at nominal level 0.05 using BRASS. SNPS from the 38 autosomes of the domestic dog genome are depicted.

$\Phi$, and hence $\hat{\Sigma}$, is obtained prior to fitting the null model, we use it to compute $\hat{C}$. Therefore, the most computationally intensive step is to obtain the eigenvectors of $\hat{\Psi}$ (the estimate for $\Psi_0$ in (11)), which involves a time complexity of $O(n^3)$. By using the singular value decomposition of the matrix $W$ in (11), which involves a time complexity of $O(nk^2)$ (assuming the number of

**Table 1. Top association signals for ED.**

| SNP | Chr | Position | MAF | Single SNP p-value | Genomewide p-value |
|---|---|---|---|---|---|
| rs9000666 | 26 | 16554631 | .18 | $1.0 \times 10^{-7}$ | .010 |
| rs21895578 | 1 | 77938330 | .17 | $1.5 \times 10^{-7}$ | .014 |
| rs23910667 | 35 | 7272763 | .08 | $3.6 \times 10^{-7}$ | .033 |

covariates $k < n$), we are able to obtain the eigenvectors of $\hat{\mathbf{\Psi}}$ without having to construct it and compute its eigen decomposition. Past this step, the additional cost for generating $L$ replicates is $O(n^2 L)$; this computation can easily be parallelized to generate sets of traits replicates independently.

BRASS is implemented in a freely downloadable software package at https://github.com/joellesophya/BRASS. We report run times for BRASS in simulated data. Using a single processor on a machine with 6 core Intel Xeon 3.50 GHz CPUs and 32 GB RAM, on simulated data with 1,000, 5,000 and 10,000 individuals, it takes 1.6 s, 1 min and 4.6 min (275 s), respectively, to generate 1,000 trait replicates. As the use of BRASS will vary based on the context (e.g. multiple testing correction, region-based association testing), the computational burden involved in comparing the statistic between the observed trait and its replicates will vary based on what statistic is being considered.

## Dryad DOI

https://doi.org/10.5061/dryad.266k4 [32]

## Discussion

In genetic studies of binary traits, it may often be of interest to assess significance of a test statistic whose distribution is unknown or not well-approximated, or to assess significance of the maximum of many correlated tests. Examples could include test statistics based on machine learning methods [1], rare variant tests [3], test statistics that involve data-adaptive weights [4], and assessment of genomewide significance [5] in a variety of situations such as inclusion of admixed individuals in GWAS [6]. Permutation tests are an attractive and popular approach, but in the presence of population structure, cryptic relatedness and/or family structure, which are all common sources of confounding in genetic association studies, ordinary permutation tests are usually not appropriate as they fail to retain the structure present in the data.

We propose BRASS, a novel permutation-based resampling procedure for generating binary trait replicates in samples with population structure, cryptic relatedness and/or family structure. BRASS allows for covariates and ascertainment, and it accommodates a wide range of test statistics. It uses an estimating equation approach that can be viewed as a hybrid of logistic regression and linear mixed-effects model methods, which allows for use of a GRM and/or principal components or other ancestry-informative covariates to account for sample structure. BRASS relies on obtaining an invertible transformation of phenotypic residuals to achieve approximate second-order exchangeability. After permutation, new trait replicates are obtained by inverting the transformation, in order to preserve structure present in the original sample. BRASS differs from existing methods such as Naive and MVNpermute [11] in that it incorporates key features of a binary trait such as nonlinear effects of covariates and dependence of the variance on the mean, which we show improve the performance of BRASS on binary traits.

In simulations, we demonstrate the superior type 1 error control of BRASS compared to other methods. Across simulation settings, we find that BRASS maintains control of the type 1 error under varying amounts of population structure, familial relatedness, ascertainment and sources of trait model misspecification. In contrast, all 6 of the other methods we consider show significant inflation of type 1 error in multiple settings. In addition, the Naive and MVNpermute methods, which are two of the 4 methods that are based on an LMM, are excessively conservative when covariate effects are substantial. Taken together, the results show that simply accounting for the sample structure with an LMM, without incorporating key features associated with the binary nature of the trait such as nonlinear scale for covariate effects and

the dependence of the variance on the mean, is not sufficient to obtain replicates that correctly estimate the null distribution of the statistic. Secondarily, we find that when polygenic effects are important, adjusting for the correlation in the residuals prior to permutation, as is done in BRASS and MVNpermute leads to better control of the type 1 error compared to ignoring it, as is done when permuting the raw residuals from a LMM (Naive method). Furthermore, we find that the additional step of binarizing the quantitative replicates results in improvement in the accuracy of the type 1 error rate for the LMM-based methods when the polygenic component has a low impact on the trait distribution relative to covariates, but does not control the type 1 error when more structure is present in the sample.

In principle, use of a logistic mixed model would seem to be a natural and promising alternative to BRASS for fitting the binary trait data and generating appropriate replicates that preserve the correlation structure. However, for LogMM-PQL, we observe significant inflation in the type 1 error rate, with the amount of inflation increasing as the confounding due to structure present in the sample increases. This is likely explained by the use of PQL to fit the LogMM, as PQL is known to be fast but to give biased estimates in binary data [28]. However, fitting a LogMM is computationally intensive and requires a trade-off between computational time and accuracy of the approximation of the high-dimensional integral involved. Algorithms with higher accuracy than PQL (e.g. the Laplace approximation [29] or Gauss-Hermite quadrature [30]) lead to increased computational burden and do not scale well for even moderately large GWAS. In contrast, BRASS is seen to be extremely fast computationally.

We applied BRASS in the context of multiple testing correction in association mapping studies of ED and IE in domestic dogs. In the analysis of ED, we detected genomewide significant association with 3 loci, 2 of which were previously reported [27], with the 3rd locus also reaching significance based on the BRASS threshold. In the analysis of IE, we detected a 12 Mb region on chromosome 4 that reached the BRASS genome-wide significance threshold and has been previously associated with IE [27]. For the ED data analysis, breed information was available, but we chose to instead use PCs as ancestry informative covariates, due to apparent inaccuracies in the breed labeling information as well as the presence of dogs of mixed or unknown breed in the data set. Interestingly, the genome-wide p-value threshold estimated using replicates from BRASS for the ED trait in a sample of 82 breeds was about half that for the IE trait in a single breed. This is expected given the extremely different population structure between the two samples.

BRASS is designed as a very flexible tool that can be used in a range of situations in which replicates of correlated binary data are needed. Because BRASS generates replicates of the phenotype, it can be used with a variety of genetic predictors. For example, it could be directly applied to association testing of haplotypes, to association testing of variants on sex chromosomes, or to tests of GxE or GxG interaction. Although imbalanced case-control ratio can have an impact on some association testing methods, it would not be expected to have any particular impact on generation of BRASS replicates, because (1) the appropriate modeling of the connection between the mean and the variance for binary data means the method does not break down for small or large probabilities, in contrast to an LMM, and (2) the asymptotic approximations used for inference in logistic regression are not needed in BRASS. In our simulation results, for the settings with imbalanced case-control ratio (Figs 1, 2, 4, 5), there was no noticeable impact. In our simulations, we have considered various types of model misspecification and demonstrated robustness of BRASS for association testing. We note that certain applications such as testing of interaction have been shown to be more sensitive to model misspecification than association testing is [25], so additional sensitivity analyses may be needed when performing interaction analyses, regardless of the method used to assess significance.

Depending on the application, one can consider a different approach to capture the structure present in the sample, as the model used by BRASS can accommodate both covariates and random effects. In the simulation studies and data analysis, we chose to partition the relatedness into random and fixed effects and include both in our model. A previously proposed method, PC-Air [31], obtains PCs from a derived subset of mutually unrelated subjects that are representative of the ancestral diversity present in the sample, so as to ensure that the top PCs will only capture distant genetic relatedness (i.e. population structure). Such methods could easily be used to obtain the population structure information input to BRASS, if desired.

BRASS generates non-binary replicates, which can be used directly in some types of analyses, e.g., when the data are analyzed using a linear mixed model or using the CARAT [18] or CERAMIC [17] binary trait methods, because these methods are based on solving certain estimating equations and work well provided that the first and second moments of the data are appropriately modeled. However, analysis by ordinary logistic regression would require binary data. If binary replicates are needed and if the additive polygenic component is estimated to have a low impact on the trait distribution relative to the covariates, then we would recommend either BRASS$_{mod}$ or MVNpermute$_{mod}$ which both provided good type 1 error control in our simulations in those cases.

## Supporting information

**S1 Text. Detailed methods and additional results.** Detailed description of the methods, including simulation of genotypes, traits and covariates, incorporating sample structure in the null model, adaptive resampling procedure, and modification of CARAT. Additional results consist of p-values for IE and ED loci from other permutation-based methods.
(PDF)

**S1 Fig. Three-generation pedigree used in the simulation studies.**
(PDF)

**S2 Fig. Empirical Type 1 Error Rates Omitting a Relevant Covariate at Nominal Level 0.01.**
(PDF)

**S1 Table. Differences between the seven resampling methods compared in the simulation studies.**
(PDF)

## Author Contributions

**Conceptualization:** Joelle Mbatchou, Mark Abney, Mary Sara McPeek.

**Formal analysis:** Joelle Mbatchou, Mary Sara McPeek.

**Funding acquisition:** Mary Sara McPeek.

**Investigation:** Joelle Mbatchou, Mary Sara McPeek.

**Methodology:** Joelle Mbatchou, Mary Sara McPeek.

**Project administration:** Mark Abney, Mary Sara McPeek.

**Resources:** Mark Abney, Mary Sara McPeek.

**Software:** Joelle Mbatchou, Mark Abney, Mary Sara McPeek.

**Supervision:** Mark Abney, Mary Sara McPeek.

**Validation:** Joelle Mbatchou.

**Visualization:** Joelle Mbatchou, Mary Sara McPeek.

**Writing – original draft:** Joelle Mbatchou, Mary Sara McPeek.

**Writing – review & editing:** Joelle Mbatchou, Mark Abney, Mary Sara McPeek.

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
