## [Decision Letter · Decision Letter 0]

28 May 2023

Dear Dr McPeek,

Thank you very much for submitting your Methods entitled 'BRASS: Permutation methods for binary traits in genetic association studies with structured samples' to PLOS Genetics.

Both reviewers have evaluated the manuscript and found the study interesting and valuable. They made however consistent comments in terms of challenging the simulations studies: more complex population structure and a simulation that combines all the complications that were considered separately in your initial set of simulations. We do not see the addition of more complex population structure as essential, though it may be interesting and relatively straightforward to investigate that point. However, the point about combining ascertainment, imbalanced case-control ration, missing covariates and phenotype misspecification is an interesting one, and should be relatively easy to implement given that most of these features have been considered separately. No analytical strategy is perfect and this combination may break the permutation scheme, but that would be interesting to report.

If you decide to revise the manuscript for further consideration at PLOS Genetics, please aim to resubmit within the next 60 days, unless it will take extra time to address the concerns of the reviewers, in which case we would appreciate an expected resubmission date by email to plosgenetics@plos.org.

We are sorry that we cannot be more positive about your manuscript at this stage. Please do not hesitate to contact us if you have any concerns or questions.

Yours sincerely,

Vincent Plagnol

Academic Editor

PLOS Genetics

David Balding

Section Editor

PLOS Genetics

Reviewer's **Comments to the Authors:**

Reviewer #1: This paper described novel permutation methods to assess genome-wide significance in structured population. This is a timely topic that has not been well addressed in the literature. The authors have described relevant work in background and proposed a new residual-based permutation method in binary trait association mapping. Simulation studies were performed to compare their methods with other permutation methods and demonstrated well controlled type I error. The proposed method was applied to the dog GWAS study and identified a new association. My specific comments are as follows:

1. In the simulation study, the genotype data were generated using 2 sub-population model. How does BRASS perform with more complex population structure, such as more sub-populations and admixture?

2. In the real data analysis with dog GWAS, the original paper described more than two phenotypes. What is the motivation that this study focused on ED and IE? It would be beneficial for readers to see the performance of BRASS across more phenotypes in this dataset. Another option could be the demonstration of BRASS in multiple studies.

Reviewer #2: The review report is uploaded as an attachment.

**Have all data underlying the figures and results presented in the manuscript been provided?**

Reviewer #1: Yes

Reviewer #2: Yes

PLOS authors have the option to publish the peer review history of their article (what does this mean?). If published, this will include your full peer review and any attached files.

Reviewer #1: No

Reviewer #2: No

---

## [Decision Letter · Decision Letter 1]

16 Oct 2023

Dear Dr McPeek,

We are pleased to inform you that your manuscript entitled "BRASS: Permutation methods for binary traits in genetic association studies with structured samples" has been editorially accepted for publication in PLOS Genetics. Congratulations!

Yours sincerely,

Vincent Plagnol

Academic Editor

PLOS Genetics

David Balding

Section Editor

PLOS Genetics

Reviewer's **Comments to the Authors:**

Reviewer #1: Although the authors did not consider more sub-populations, they expanded their simulations to address ascertainment, imbalanced, case-control ratio, missing covariates and phenotype misspecification. For data analysis, the authors justified their choices of two phenotypes in the dog GWAS, which I'm fine with.

Reviewer #2: Thank you to the authors for addressing my previous comments. I have no further comments on the revised paper.

**Have all data underlying the figures and results presented in the manuscript been provided?**

Reviewer #1: Yes

Reviewer #2: Yes

PLOS authors have the option to publish the peer review history of their article (what does this mean?). If published, this will include your full peer review and any attached files.

Reviewer #1: No

Reviewer #2: No

**Data Deposition**

http://datadryad.org/submit?journalID=pgenetics&manu=PGENETICS-D-23-00252R1

**Press Queries**

---

## [Editor Report · Acceptance letter]

30 Oct 2023

PGENETICS-D-23-00252R1 

BRASS: Permutation methods for binary traits in genetic association studies with structured samples 

Dear Dr McPeek, 

We are pleased to inform you that your manuscript entitled "BRASS: Permutation methods for binary traits in genetic association studies with structured samples" has been formally accepted for publication in PLOS Genetics! Your manuscript is now with our production department and you will be notified of the publication date in due course.

With kind regards,

Lilla Horvath

PLOS Genetics

On behalf of:
